# Reviews and syntheses: The promise of big diverse soil data, moving current practices towards future potential

Katherine E. O. Todd-Brown[1], Rose Z. Abramoff[2, 3], Jeffrey Beem-Miller[4], Hava K. Blair[5], Stevan Earl[6], Kristen J. Frederick[1], Daniel R. Fuka[7], Mario Guevara Santamaria[8], Jennifer W. Harden[9], Katherine Heckman[10], Lillian J. Haren[1], James R. Holmquist[11], Alison M. Hoyt[9], David H. Klinges[12], David S. LeBauer[13], Avni Malhotra[9, 14], Shelby C. McClelland[15], Lucas E. Nave[16], Katherine S. Rocci[17], Sean M. Schaeffer[18], Shane Stoner[4, 19], Natasja van Gestel[20], Sophie F. von Fromm[4, 19], and Marisa L. Younger[1]

[1]Department of Environmental Engineering Science, University of Florida, Gainesville, Florida, USA
[2]Laboratoire des Sciences du Climat et de l'Environnement, Gif-sur-Yvette, France
[3]Environmental Sciences Division, Oak Ridge National Laboratory, Oak Ridge, Tennessee, USA
[4]Max Planck Institute for Biogeochemistry, Jena, Germany
[5]Department of Soil, Water, and Climate, University of Minnesota, St Paul, MN, USA
[6]Global Institute of Sustainability and Innovation, Arizona State University, Tempe, AZ, USA
[7]Virginia Polytechnic Institute and State University, Blacksburg, VA, USA
[8]Centro de Geociencias, Universidad Nacional Autónoma de México, Juriquilla, Querétaro, Mexico
[9]Department of Earth System Science, Stanford University, Stanford, CA, USA
[10]Northern Research Station, USDA Forest Service, Houghton, MI, USA
[11]Smithsonian Environmental Research Center, Edgewater, Maryland, USA
[12]School of Natural Resources and Environment, University of Florida, Gainesville, Florida, USA
[13]Arizona Experiment Station, College of Agriculture and Life Sciences, University of Arizona, Tuscon, AZ, USA
[14]Department of Geography, University of Zürich, Zürich, Switzerland
[15]Department of Soil and Crop Sciences, Graduate Degree Program in Ecology, Colorado State University, Fort Collins, CO, USA
[16]Biological Station and Dept. of Ecology and Evolutionary Biology, University of Michigan, Pellston, MI, USA
[17]Natural Resource Ecology Laboratory, Department of Soil and Crop Sciences, Graduate Degree Program in Ecology, Colorado State University, Fort Collins, CO, USA
[18]Biosystems Engineering and Soil Science Department, University of Tennessee, Knoxville, TN, USA
[19]Department of Environmental Systems Science, ETH Zürich, Zürich, Switzerland
[20]Department of Biological Sciences & TTU Climate Center, Texas Tech University, Lubbock, Texas, USA

**Correspondence:** K Todd-Brown (ktoddbrown@ufl.edu)

**Abstract.**

In the age of big data, soil data are more available and richer than ever, but -outside of a few large soil survey resources- remain largely unusable for informing soil management and understanding Earth system processes beyond of the original

study. Data science has promised a fully reusable research pipeline where data from past studies are used to contextualize new findings and reanalyzed for new insight. Yet synthesis projects encounter challenges at all steps of the data reuse pipeline, including unavailable data, labor-intensive transcription of datasets, incomplete metadata, and a lack of communication between collaborators. Here, using insights from a diversity of soil, data and climate scientists, we summarize current practices in soil data synthesis across all stages of database creation: availability, input, harmonization, curation, and publication. We then suggest new soil-focused semantic tools to improve existing data pipelines, such as ontologies, vocabulary lists, and community practices. Our goal is to provide the soil data community with an overview of current practices in soil data and where we need to go to fully leverage big data to solve soil problems in the next century.

## 1   Introduction

Soils host a myriad of life forms, from viruses to macro fauna, that govern important ecosystem functions, such as nutrient cycling, water retention, pollutant remediation and carbon sequestration. Soils play a critical role in human existence, both as a habitat and as a source of food, fuel and fiber. Additionally, soils contain the largest pool of actively cycling carbon (Ciais et al., 2013) and can contribute to climate change mitigation through soil carbon sequestration management practices (Smith et al., 2020; Fuss et al., 2018). The health of this critical resource is under threat from mismanaged land-use changes, leading to reductions in soil organic matter and changes in soil structure that have led to wind and water erosion (Lehmann et al., 2020). Because of the soil's diverse role in food security, climate change, society, and ecosystem functioning, soils have been the central focus of many studies resulting in a wealth of data.

Soil data are as diverse as soils themselves, reflecting interactions of biological, chemical, hydrological and biophysical processes. Soil data uses include a broad range of applications such as ecology, biogeochemistry (Iversen et al., 2017; Wieder et al., 2021b), soil engineering, soil taxonomy and classification, geochemistry (Nave et al., 2016; Hengl et al., 2017; Lawrence et al., 2020), micrometeorology (Cheah et al., 2018), agronomy (Lyons et al., 2020), and geomorphology. Datasets, defined as "a collection of scientific data including primary data and metadata organized and formatted for a particular purpose" (Löffler et al., 2021) are assembled by an equally diverse range of organizations. These organizations include government agencies, academic collaborations, nongovernmental organizations, and industry, reflecting a wide range of generators and users including farmers, land managers, students, technicians, scientists, and policy makers. Recent efforts have summarized the availability of soil data, database types, their strengths and weaknesses, and how research and data networks can help solve key soil-related research and societal problems (Harden et al., 2018; Malhotra et al., 2019; Weintraub et al., 2019). These soil databases are a key tool for a wide variety of users, such as the scientific community, land managers and policy makers.

The creation of multi-sourced data products requires unification of data, through the process of harmonization. "Database", for the purposes of this paper, is defined as a synthetic collection of soil observations, including location, physicochemical properties, units, and methods. Databases can consist of data spanning multiple curation levels - from raw, to processed, to gap-filled - and are often assembled from different sources including collaborating investigators' datasets, disparate intermediate datasets, or large existing databases. Further, these databases are often cross-referenced, for example, by integrating geo-

referenced meteorological data or remote sensing products like net primary production maps. "Harmonization" of data refers to the integration process and can include activities like restructuring data tables. A number of databases have been compiled in soils around specific themes or measurement types including: soil carbon and nitrogen (Worldwide soil carbon and nitrogen data (Zinke et al., 1998), International Soil Carbon Network database (Nave et al., 2016)), field based soil respiration (Soil Respiration Data base (Bond-Lamberty and Thomson, 2010)), lab-based heterotrophic respiration (Soil Incubation Database (Schädel et al., 2020)), Soil radiocarbon (International Soil Radiocarbon Database (Lawrence et al., 2020)), and coastal soils (Coastal Carbon Research Coordination Network Database (Holmquist et al., 2018 - 2021)) (see Section A). The soil resources and data products that make up the World Soil Information Service (WoSIS) curated by International Soil Reference and Information Centre (ISRIC) (Batjes et al., 2020) is an example of how soil data feed into larger products. After archival on ISRIC servers, datasets from individual providers are incorporated in the the World Soil Information Servers workflow of mapping diverse data contributions in to a standard data model, harmonization and distribution (including both databases and derived data products, such as SoilGrids (Hengl et al., 2017; Hengl et al., 2014)). Harmonized databases can be a powerful validation tool for Earth System models as well as other modeling and estimation efforts and to construct these databases the underlying data should adhere to best practices.

The FAIR (Findable, Accessible, Interoperable, and Reusable) Data Principles have become a popular shorthand for best practices in scientific data management and stewardship across scientific domains (Wilkinson et al., 2016). Much of the previous work on FAIR principles has focused on data access (though notable counter examples include Goble et al. (2020)), which can be difficult if data relies on an 'available on request' note included at the end of publications (Savage and Vickers, 2009; Vines et al., 2014). Indeed previous research has identified challenges with educating and motivating data providers to publish their datasets (Wolkovich et al., 2012). It is important to note here that FAIR does not always mean open freely reusable data. Indeed the FAIR Data motto makes this difference: "as open as possible, as closed as necessary", and this becomes particularly important for data that has possible economic impact (Luque, 2019). While data access has improved as funding agencies requiring data management plans that include public archiving, more widely available data does not ensure that they are interoperable and reusable. Many of the challenges and best practices we highlight below address this latter half of FAIR data - interoperability and reusability - and reflect the diversity of soil data generators, users, and stakeholders. In this work, we hope to guide prospective soil data providers, users, and synthesizers by highlighting both challenges and examples of working toward FAIR data within the soil science community.

While it is not desirable nor tenable to force all soil data to adhere to a single data template, there are shared strategies, semantic tools, community practices, and protocols for data harmonization and integration efforts that would increase transparency and decrease initialization effort of these projects. We set out in this paper to summarize current practices in soil data management and database harmonization, as well as suggest new semantic and community tools for the future of soil harmonization. The approach and issues outlined in this paper are undoubtedly not unique to soils and are relevant to a wide range of scientific data, particularly environmental data. However we present this as a case study of soil specific database construction. We offer this summary from the collective viewpoint of over two dozen researchers across North America and Europe, working on a broad array of funded and unfunded projects to construct independent harmonized soil databases. With our perspective

and definitions for context, our specific objectives were to define common and aspirational practices in harmonized database development, and envision next steps to improve reproducibility and transparency of harmonized database development for soils.

## 2 Current database pipeline

A database pipeline is a series of steps that brings data from an initial format to a harmonized database, with steps including: discovery, acquisition, input and harmonization, curation, gap-filling and pruning, and publication of the database (Figure 1). While these general steps are fairly easy to articulate, their specific implementation varies and is often time consuming. Discovery, access, and "wrangling" can exceed 80% of the effort required for a new scientific discovery (Lohr, 2014; Beno et al., 2017). The extension of this pipeline to create a generalizable solution adds an additional level of difficulty (Furche et al., 2016), but can be essential to ensure that both the data and workflows are FAIR (Goble et al., 2020). For brevity, we assume that datasets of interest have already been identified. However, searchability of data repositories is an active area of research (Pampel et al., 2013; Löffler et al., 2021). We briefly review several common approaches to data acquisition, harmonization, curation, and publication.

### 2.1 Availability

Reproducible analysis is fundamental to robust science and data analysis. Naively, a newcomer to the field of 21st century science might be forgiven the assumption that a published peer-reviewed journal article would, by default, also be accompanied by a published dataset in a machine readable format. In the authors' experiences, this is uncommon for a number of reasons.

While some peer-reviewed journals and funding organizations require the data to be deposited in a trusted repository that supports the FAIR principles (Fox et al., 2021), confirming that data meet these high standards is often overlooked during the review process. Indeed, there is often confusion in the field as to what exactly such 'high standards' are and whom is responsible for ensuring these standards are met. To complicate matters, key contextual data for one study may be mostly irrelevant for a second. Anticipating these contextual data needs is challenging and leads many data providers who would otherwise support data sharing to become frustrated with the existing guidance (Couture et al., 2018). This is not to say that archiving data for the purpose of meeting funder requirements or reproducing the associated analysis can not be useful in and of itself; however this does not automatically lend the data to integration in a database.

On the data aggregation side, many data aggregators are challenged by unclear data documentation and metadata. This ambiguity can lead to a range of interactions between data providers and data aggregators that can vary from no contact (e.g. left the field due to career changes, retirement, death, or are unwilling to interact) to high-contact (data providers collaborate with data aggregators to fill out a harmonized template). Intermediates along this gradient could include reaching out to data providers to confirm variable ranges, addressing possible errors, requesting specific unpublished measurements, or clarification of ambiguous descriptions. Data providers have unique knowledge about their systems and can be instrumental in expanding or

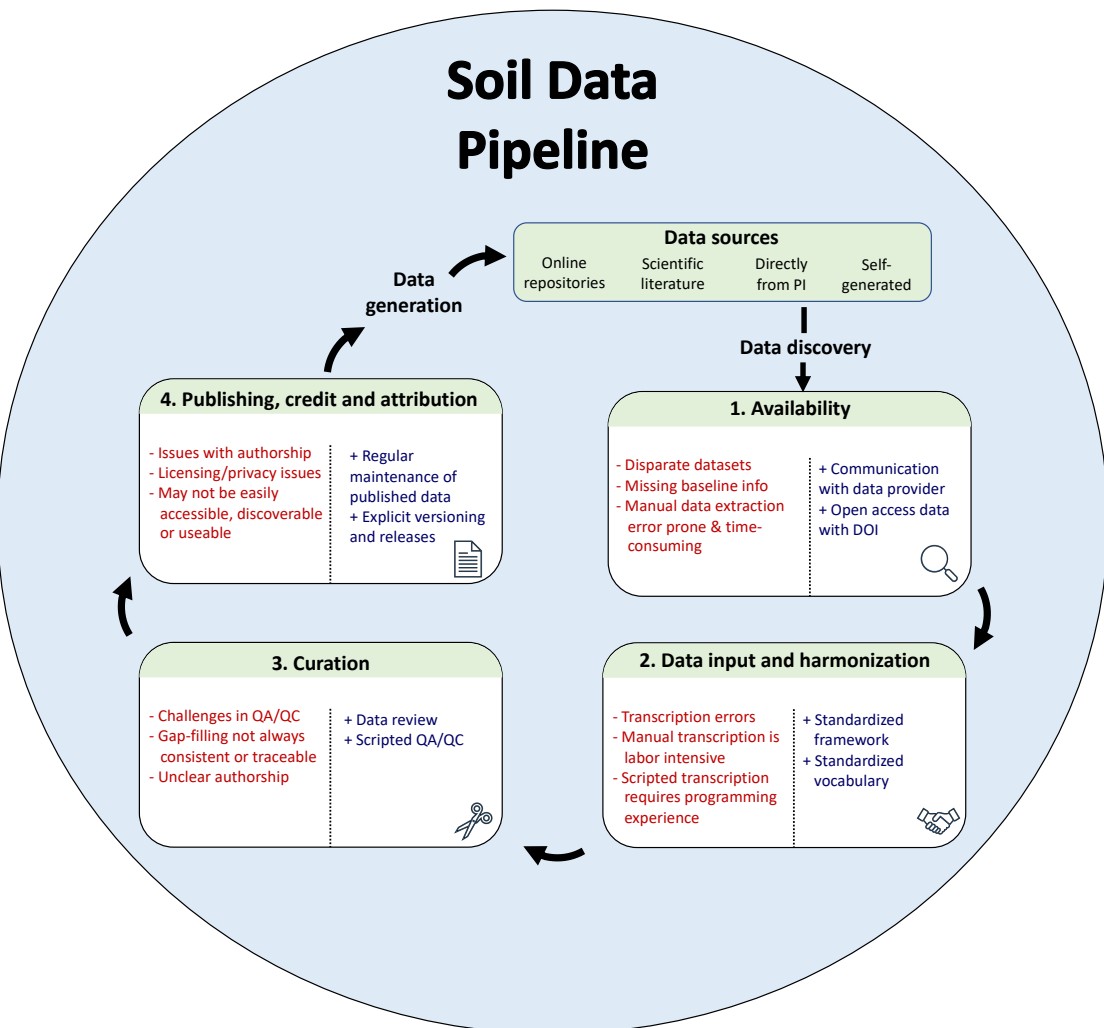

**Figure 1.** Database pipeline with pain points (indicated by "-") and suggestions for improvement (indicated by "+") to conform more closely to FAIR Data Principles. Data sources can be diverse, including published (online repositories and scientific literature) and unpublished sources (direct from principal investigator; PI). After these sources have been discovered, the data must be accessed, harmonized according to a standard format or data model (internal to the project or a community-driven standard). The aggregated data must then be curated before ultimately being published for reuse.

modifying the scope of the resulting database analysis. Data-centered collaborations can lead to new communities of practice and better science (see Future recommendations section).

In addition to these benefits there are also trade-offs with contacting the original data providers. Acknowledgement and level of visibility of original data contributions remain an open question. Data providers may expect to be listed as co-authors upon reuse of their data, in recognition of their past effort collecting the data, despite having limited or no engagement in the reuse project. This is often frustrating to a data aggregator who, in turn, is left with an ever expanding list of coauthors with varying levels of involvement (see Section 2.4). Differing expectations on acknowledgements can lead to conflict and result in a lack of trust in the community (Longo and Drazen, 2016).

Direct collaboration between data providers and data aggregators is a critical relationship to nurture, but exists in a broader context of good data stewardship practices. The FAIR Principles for good data practices have been thoughtfully extended via TRUST and CARE. The TRUST Principles (Transparency; Responsibility; User focus; Sustainability; Technology) articulate key features for trustworthy digital repositories, which are essential for preserving data access and reuse over time (Lin et al., 2020). The CARE Principles for Indigenous Data Governance (Collective benefit, Authority to control; Responsibility; Ethics) position decisions related to data management and reuse in the context of Indigenous culture and knowledge systems, highlighting actions that ultimately support community-based data sovereignty (Carroll et al., 2020). As the community continues to discuss shared tenets of good data governance, it is becoming increasingly clear that 'just put it on a repository' is only the beginning.

## 2.2 Data input and harmonization

Data formats typically reflect the purposes of the study; thus examining data to address a different question or context often requires a different data format. Harmonizing one data contribution with a broader collection entails merging or breaking apart data tables, renaming columns, and occasionally converting units of observation (see Section 2.3 below). Translation of the data table structure (often called the data model) is typically handled one of three ways: manual template transcription, scripted template transcription, or keyed translation. All three methods have justifications for use, and data aggregators may choose to turn from one method to another as a database grows. Regardless of which method is used, one of the primary goals is to maintain data provenance that allows each data point to be traced to an original study or author.

Manual transcription is the most common method, and typically entails data taken from the original source and entered into a common template by either the data provider or aggregator (see Sections A1 and A2). Asking the data provider to fill out these data templates is often identified as a major hurdle to contribution, yet data aggregators may be unfamiliar with the data provided and thus capture an incomplete or incorrect translation of the original data into the new format without the help of the data provider. Regardless of who fills out the template, manual transcription of data is error prone. In some cases this is unavoidable when the data are not available in a machine-readable format. There are a number of software that allow for data extraction from figures (i.e. Web Plot Digitizer (Rohatgi, 2021), Data Thief (Tummers, 2006), and metaDigitise (Pick et al., 2018)) and tables (pdftools (Jeroen Ooms, 2021), tabula (Aristarán et al., 2012 - 2020)), that can reduce human error in transcribing these machine-hostile formats. Despite its flaws, manual transcription is flexible and easy to set up, making it a frequent choice for data aggregation studies with a tight timeline.

An alternative approach to manual transcription is scripted transcription. Scripted template transcription involves writing a computer program, customized to the specific data being ingested, to reformat the data tables and column names to match a target data standard or template (see Section A3). This approach requires familiarity with both soil science (to understand the measurements) and programming (to write the scripts). In practice, the authors find such a skill combination unusual for any individual researcher, necessitating the use of interdisciplinary teams and adding organizational complexity. The codebase can also become unwieldy if written on a case-by-case basis for each input dataset. These costs are countered by an increase in accuracy, transparency, and reproducibility when compared with manual transcription.

Keyed translation is the most general approach and, as a result, requires the most extensive informatics work. Keyed translation is related to scripted transcription but uses a dictionary to define relationships between the input data format and target data format; for example mapping Column 1 to Column A or Table A to Table B. Keyed translation combines metadata about each dataset with a generalized conversion script to generate a harmonized database (see A4). While this approach has the most explicit need for clear semantic resources, these are also essential for creating effective manual transcription templates and protocols. Such a generalized approach can be more easily extended to expand the number of data sources. However there is currently no broadly agreed upon annotation vocabulary in the field of soil science, making it necessary to annotate each dataset individually within each project. In addition, the computational expertise needed for this approach is the highest of the three outlined here. While we feel this holds great promise for future studies, this is an uncommon approach due to these challenges.

## 2.3   Curation

Data that are in an integrated database often still need to be curated to ensure accuracy, convert units, address missing data or gaps, and reduce or aggregate data to derive relevant data products. While scripts are often used extensively at this phase, expert interpretation and review is a critical component. Finally, reuse of databases often requires a repeat of this curation phase – what may be appropriate for one question or purpose may not be appropriate for another.

Scripting is heavily utilized to augment expert review to quality control data. These scripts both automate and document quality control criteria, however setting those criteria often requires an extensive knowledge of the system and measurement methods. For example, the ISRaD database has an automated quality control protocol accessible via web interface which ensures that values are within reasonable ranges, and checks that records and critical metadata are appropriately linked across database tables (Lawrence et al., 2020). Following this initial filter, a manual individual 'expert review' is conducted by a trained ISRaD volunteer (see Section A2). These extensive quality control procedures require time and diverse expertise, making them unattractive for many open source database projects without broad recognition of service by the field.

Gap-filling expands data coverage, and can include a number of strategies to fill in missing data at both the layer or horizon level as well as the profile or site level. For example, expanding the environmental context for a particular soil sample location, such as extracting net primary productivity and land use classification from satellite products, is one example of gap-filling soil-relevant observations that may not have been collected at the time of the soil sampling. Strategies for gap-filling more broadly include linear interpolation, pedotransfer functions, georeferenced data extraction, or more sophisticated machine learning

algorithms. Given the wide variability in gap-filling practices and objectives, these methods must be extensively documented, clearly state use case restrictions, and estimate uncertainties.

Finally, a common step is to remove unnecessary data from the data product via pruning. Pruning removes samples based on location or type of measurement after database compilation to cater to specific data user needs and reduce the size of the data product. Both pruning and gap-filling highlights the importance of maintaining an intermediary harmonized database as well as the final data product both to preserve the original contextual data and reuse of that data for alternative projects.

## 2.4 Publication, credit and attributions

Authorship issues are common in data aggregation projects due to unclear expectations and conflicting conventions across what are often large teams of collaborators. Indeed authorship issues are common in large collaborative projects (Cooke and Hilton, 2015). This can be mitigated with an inclusive co-author list to include both data providers, aggregators, and re-analysis teams, but requires significant project management and organizational overhead. As always with larger team science, we highly recommend a formal authorship policy prior to beginning database compilation including what the role of each contributor is and who are in lead manuscript positions, listed as co-authors, and listed in acknowledgements. While it can be tempting in a data aggregation project to fall back on what is accessible and legally entitled, we strongly feel that more inclusive projects build trust within the scientific community, leading to better data interpretation and seeding future collaborations.

Related to issues of credit and co-authorship described above are issues with data licenses. Similar to manuscripts, data are often released under a specific legal license with requested reuse considerations, which may hinder the inclusion of otherwise 'public' data in a data synthesis. There is a tension in choosing between an adequately restrictive license, which can help ensure that a specific project and data providers are given credit, and a permissive license, which can increase data reuse. The Creative Commons provides a framework to examine these considerations but there are many other standard and custom licenses. The most permissive is CC-0 or public domain license that puts no restriction on data use. A 'By-Acknowledgment' (BY) rider requires that the original data source be acknowledged in the derivative product in some way (sometimes this acknowledgement is specified, sometimes not). A 'Non-Commercial Use' clause restricts the sale of the data for commercial purposes. Finally a 'Share-Alike' or copy-left clause says that the data may be reused if it is released under the same license. A 'CC-BY' license is probably the closest to the traditional academic practice of research citation and many scientific repositories including the Environmental Data Initiative and Pangaea encourage data providers to select this option.

In all cases database creation does not have to be a single push, but is ideally part of an ongoing synthesis effort, leading to the need for database versioning. The COSORE database is an example for such an approach (Bond-Lamberty et al., 2020). After each major change (release) the database receives a new doi and is permanently archived on a repository. This allows maximal transparency, allowing data users to reproduce an analysis from a given version and making it easy to find the newest version of the database.

## 3  Future of soil data

The hope of big open data is to have any data collected at any time anywhere in the world at the tip of your fingers (see Section B). To reach this hope it's important not just to work with large volumes of data but also diverse observations and measurements in a way that is trusted. For soil science the potential for long-term (multi-decadal) understanding is particularly exciting. Long-temporal coverage of soil data could lead to a better understanding of soil carbon sequestration potential to mitigate climate change, or better management of soils for crops. How do we attain these above futures, where data reuse is equally as valued as data production?

We recommend implementing a core set of measurements and processes to facilitate soil data reuse. The recommendations in section 3.1 are aimed at researchers collecting soil data who wish to ensure the long-term value and reusability of their datasets. These recommendations are also relevant for journals and peer reviewers of soil science research as a short checklist of key details that should be reported or addressed. Section 3.2 outlines recommendations for researchers who wish to participate in the data harmonization process. These recommendations encompass both technical and social considerations for data harmonization efforts and focus on what can be done right now to further soil data exchange.

### 3.1  What to measure and report?

Soils are inherently rooted in time and space, making high resolution spatial and temporal information (including sampling date, latitude, longitude, geographic datum, and depth of sample) critical for building context and data reuse. Data providers will often ask 'what should I measure' to be relevant to data aggregation efforts, and there are efforts to provide such guidance (Billings et al., 2021). We have chosen instead to focus on critical temporal-location information to allow data to be expanded, contextualized, and annotated. The issue is not that researchers do not know how to record this information, but rather conflicting objectives may prevent its recording.

Geospatial metadata may present a privacy concern, for example when the soil measurements are tied to the economic valuation of the land as in agricultural systems. For data collected on privately owned land, such as on-farm research and observations, researchers may not be at liberty to release detailed location information publicly in order to protect landowner privacy (Richardson et al., 2015). There are efforts to bridge data sharing and data privacy. For example, the platform under development by the International Agroinformatics Alliance will integrate secure data storage, granular data permissions, and options to register privately hosted data to facilitate data discovery and sharing while protecting privacy (Gustafson et al., 2017). Clearly this is an ongoing discussion that will require more research and conversations with stakeholders.

The advantages of high precision geolocations are significant; and regardless of the precision, the level of uncertainty in the provided geolocation is critical and often missing in archived datasets. Location information enables soil data to be joined with the growing number of gridded global datasets that can provide key contextual information for interpretation and modeling. While there are privacy concerns in some locations, not reporting the location of a sample collection should be the exception and not the rule, especially in publicly funded research data.

In addition to location, sample depth is also critical due to the variation of soil properties and processes with depth. Unfortunately over 60% of studies fail to report measurements of sample depth (i.e. layer defining upper and lower bounds) associated with soil data (Yost and Hartemink, 2020). This is particularly critical to advancing our understanding of deeper soil properties and functions, but also relevant for the effects of surface tillage and grazing on managed lands.

Soils are temporally dynamic and time of collection can provide key insights into decadal level changes in soils. Soils change over time owing to pedogenesis, historical land use and increasingly, global climate change (Tugel et al., 2005; Richter et al., 2011; Ellis, 2011; Harden et al., 2018). Recording the time of collection for modern datasets can produce valuable returns in future reanalysis and depending on the measurement the exact resolution will vary (for example, parent material may just need the decade of collection while soil respiration may need a minute resolution). Older datasets, consisting of historic measurements and archived samples, are increasingly valuable to track soil responses to global change. Such datasets can provide a window into the dynamics of how soil properties change, and should be a high priority for data rescue and documentation.

Long-term (decadal scale) soil records provide valuable information for the study of global change and land management, and therefore sites associated with older observations should be prioritized for re-survey efforts (Hawkins et al., 2013). Often referred to as data 'rescuing', there is an staggering potential of decadal-scale data sitting in labs across the world but best practices on how to use this data are lacking. Decadal data are often associated with a particular researcher or group and may represent an entire career of data collection that likely have been reformatted multiple times across several generations of storage systems and lab staff and may even be on analog storage and not digitized. Augmenting (or extending) rescuing data from a single group is resurveying older sites and conducting structured interviews of personnel to enrich metadata of prior observations (Karasti et al., 2006). In the opinion of the authors, data rescue efforts are an underutilized resource in the field.

By adopting these recommendations to record geolocation, depth of sample, and collection date we can greatly increase the value of soil data, extending the measurement reusability for future analysis.

## 3.2   How to harmonize?

We touched on several common approaches to data harmonization in this paper. Often driven by a single research question or objective, data harmonization has historically been a laborious process carried out by a single or small group of researchers for a specific project. Based on our experiences in various harmonization projects, we propose a more community-centered approach moving forward, founded on the principles of open and transparent science. Outputs from these groups should include semantic tools like ontologies and shared vocabulary lists with clear and transparent governance, as well as a new community-centered approach to the practice of data harmonization and the resulting databases.

The uses of soil databases for research context are varied (for example: Earth system model benchmarking (Collier et al., 2018)) but there are other private economic impacts of having soil data available. Soil health metrics in public databases could impact land fertility evaluation and there is increasing interest in soil carbon data by carbon markets for offsetting $CO_2$ emissions. As mentioned in Section 3.1, specific information on nutrient and water retention of a soil can make it more or less valuable leading to landowner reluctance to release data. More recently an increasing interest in generating carbon offsets

by increasing soil carbon sequestration has lead to a proliferation of new venture corporations that either generate new or use available soil data in order to define land management practices (e.g. IndigoAg, CIBO Technologies, Seqana, Regrow, Nori, LoamBio). Industry companies generally treat data that they collect or process as part of their intellectual property, which is pent private. While there is clearly scientific value in these data, it is unclear how researchers, landholders, and private companies will negotiate the use and integration of these data into research output. Nonetheless, privately held data would also

benefit from and contribute to connection with community developed tools.

### 3.2.1 Community tools

There is an understandable tendency by many scientists towards data model standards (for example Rüegg et al. (2014) and as proposed for soil respiration data (Bond-Lamberty et al., 2021)). If all soil data adhered to a common template, or data model, with uniform tables and column names, then it would be trivial to append the data from one study with the data from a second.

Unfortunately, due to the diversity of soil types and methodologies, as well as ever evolving measurement technologies, we feel that this is impractical for soil research data, although several valiant efforts are underway to do this (Nave et al., 2016; Lawrence et al., 2020). In practice, researchers will continue to develop their own data tables and internal conventions that make sense for their experimental structure, location, and measurement type. However, semantic tools and standards as well as software development practices like versioning could improve interoperability (Onerhime, 2021; Crystal-Ornelas et al., 2021).

Annotating datasets with a common vocabulary forms the theoretical backbone of all data harmonization work. Whether this is a manual copy-paste from a source data table to a common data template, or creating a thesaurus that cross-references given data columns to some internal standard name, both processes rely on a vocabulary. This digital vocabulary has roots in classical soil glossaries and lab manuals that have been printed as dictionary style references. This vocabulary could be a valuable community resource but would require ongoing engagement with the research community to remain accessible, relevant, and

up to date, in addition to addressing legacy copyright issues. Further extending this vocabulary into an ontology, that captures the relationship between terms in addition to their definitions, could drive the next generation of data-driven machine learning. Community developed ontologies and vocabulary lists like ENVO (Buttigieg et al., 2016), CSDMS (CSDMS, 2019), GLOSIS (Palma et al., 2020), and CF (Hassell et al., 2017) could provide reusable resources that are currently missing and underutilized in the soil community. The soil research community as a whole needs to engage with these broader resources to ensure the

informatics reflects new developments in the understanding of soil science and measurements being made.

### 3.2.2 Community practice

Before data can talk, the data community needs to talk. Based on the experience of the authors, developing, adopting, and maintaining semantic resources is beyond the scope of any one lab or organization and requires a diverse community. In the development phase, a diverse community with a range of expertise, stakeholders, and career stages can ensure that the broadest

possible needs are being addressed. Adoption is more likely if the resource addresses the needs of the community and that community has ownership over the resource. Finally, maintaining semantic resources requires ongoing updates and revisions

as methods shift. All of this requires a new type of community, one centered on data and tools to support the interoperability of that data.

Successful data centered communities are open, transparent, diverse, and rewarding (Cooke and Hilton, 2015). They are open, in the sense that anyone can join or contribute and are empowered through educational activities to participate. Transparency ensures it is easy to contribute and to understand decision making processes. Diverse communities can draw on a wide range of skill sets, from experience in soil processes to knowledge representation. Finally, they are also rewarding, furthering members' careers through creation of tangible products (for example citations or grant dollars) and opportunities for scientific leadership and service. While there are several approaches to achieve this, one possible workflow for the establishment of a new harmonized database might look something like the following:

1. Open application period and selection of team

2. Purpose setting and establishing a common vocabulary

3. Curation and use of data product

A mix of backgrounds in software development, knowledge engineering (the design of ontologies and knowledge graphs), classical soil pedology, and specific measurement methodology (e.g. microbial characterizations, physical chemistry, and vadose zone hydrology) is required for a data harmonization project. It is often challenging to recruit such an interdisciplinary team that requires such a high level of diversity beyond traditional domain boundaries. A well-advertised open application period is a critical first step, and may be supplemented with targeted solicitations. Teams representing a range of perspectives, backgrounds, and career stages are better able to envision a wider range of end uses (and users) of soils data. These efforts are likely to result in better outcomes- improving research products and advancing the careers of people previously excluded from existing synthesis efforts, power structures, or established communities.

One of the first challenges with an interdisciplinary team is establishing agreement on goals and methods. This requires developing a shared understanding and vocabulary (e.g. through educational activities on computational tools or soils surveys and measurements). In an academic community, short-term shared purpose is most easily motivated by a synthesis paper or research question, but longer-term motivators are unclear. Data may provide a clear shared motivation but it's uses and governance processes need to be clearly identified and revisited regularly.

Sustainable creation and curation of the harmonized database is essential to create relevant data products to serve a database's specific purpose and enable future reuse to address a variety of questions. Accessing, annotating and merging the datasets is a well-established technical process once the community tools and community of practice are in place. Curation of databases could be patterned after the manuscript review process, where domain researchers review proposed database additions to ensure the accuracy of new contributions. This review process should keep the diverse needs and practices of the soil community in mind, including soil surveyors, field/lab experimentalists, and land managers. In the end, the synthesis of existing data is not the goal: it is their application to scientific problems. In that regard, successful product development from a database can encourage growth and adoption of the data resource by others.

## 4    Conclusions

Soils are the foundation for our food and fiber system as well as a significant component of the global carbon cycle. As such, information and measurements of the soil system, from hydrological conductivity to soil carbon stocks to changes in nutrient content, are a key public good for a varied group of users. However, this valuable scientific resource is currently under-utilized due to many of the issues outlined above. We suggest data use and reuse could be facilitated by addressing issues along the database construction pipeline.

We outlined database creation as a common set of steps: availability, input, harmonization, curation, and publication. While this pipeline can look different depending on the skill sets, timeline, and funding structure of the researchers involved, we summarized common pain points throughout this process, which can reduce the accuracy and usability of a database. Data collection, synthesis, and use are inherently human endeavors, and as such, breaks in this pipeline are often driven by lack of community awareness and practices.

We put forth recommendations ranging from measurement prioritization to data harmonization decisions that can help move forward community practices around soil data. We recommend that contextual information like geolocation, depth of sample, observation time, and management history all be reported with soil measurements. Soil data harmonization requires the development of new semantic tools like vocabulary lists and ontologies that are co-produced by data and soil scientists. Building the capacity to create and maintain these tools require communities of practice including: open application periods to recruit diverse participants, established goals, and clear outcomes. The creation of such communities are not an easy task, but are a needed one.

Ultimately, soil data are an invaluable resource generated and used by a diversity of groups. Given this value, we hope that the work of advancing soil information systems will increasingly be recognized and rewarded as a critical component of the research process. To achieve this, we need not only new tools and practices, but shifts in the broader incentive structure for conducting this kind of work. Our review provides a path forward to enhance community practice around soil data so that we can begin to tackle the vast array of research and management problems, and their solutions, that lay beneath our feet.

### Appendix A:  Current soil projects

Below are a series of snapshots compiled to represent the range of approaches groups currently take to aggregating soil databases. These four snapshots include a manually compiled database of field warming experiments (Section A1: Crowther et al. (2016)), a database using manual transcription combined with scripted curating of soil radiocarbon measurements (Section A2 - Lawrence et al. (2020)), a manual-scripted combination of coastal soils (Section A3 - Holmquist (2021)), and a keyed translation database of long-term observation (Section A4 - Wieder et al. (2021b, 2020)). Finally we also include a broader list of soil data collections to illustrate more generally how the field approaches these projects (Section A5).

## A1 Field-warmed soils

The template-driven approach to data harmonization is exemplified by (Crowther et al., 2016). In this study, individual researchers who collected data of interest (in this case soil field-warming manipulations) were contacted directly and invited to collaborate in a meta-analysis. A post-doc was tasked with creating a data template, and working with those collaborators to capture a representation of their study. These data were then appended into an integrated set of data tables and analyzed. By working with researchers directly, this approach captured both published and unpublished data and ensured a nuanced interpretation of the study results. This careful one-on-one approach combined with co-authorship on a high profile journal ensured that researchers were comfortable sharing data that they might have otherwise withheld from a joint publication.

One challenge with this approach is patchy secondary data. Secondary data like climate and soil physical chemical characteristics may not be critical to a small study at a single site but become fundamental to a larger cross-site analysis. Crowther et al. (2016) addressed this by extracting site-level environmental covariates from gridded geospatial files generated from global modeled predictions covering an array of climate (e.g. WorldClim (Fick and Hijmans, 2017)) and soil physiochemical characteristics (e.g. SoilGrids (Hengl et al., 2017)). Although these global predictions can be characterized by considerable uncertainty – especially at the local scale – these global products at least ensure a full set of standardized meta-data from every single location.

## A2 ISRaD

The International Soil Radiocarbon Database (ISRaD) consists of an open-source database of soil and soil-related radiocarbon data. Additionally, ISRaD provides a continually-developing library of tools for data access and manipulation based in R (Lawrence et al., 2020). Radiocarbon data production is an expensive and time-consuming process but provides unique information on longer-timescale in situ processes. In addition, a global pulse in atmospheric radiocarbon content in the 1960's provides unique analytical power for data collected in the intervening decades to constrain models.

Initial funding was provided by the USGS Powell Center and the USDA NIFA FACT program. Currently, the Max Planck Institute for Biogeochemistry provides both ongoing funding and staffing. Since 2015, over 300 studies have been compiled, though the data collection process is active and ongoing, and user-submitted data are also welcomed.

The project utilizes a standardized Excel template (based on templates designed in the International Soil Carbon Network (Nave et al., 2016)) for data ingestion which each contributor fills out and submits to a designated ISRaD coordinator. The core unit of a data entry is the "profile", which is a unique spatial AND temporal identifier. All data must be matched to a profile, which is in turn matched to a "site", and the uppermost level of "entry", which identifies the publication from which the data originates. Such hierarchy is best preserved through vetting with both an automated and human-led QA/QC process. Therefore, prior to ingestion, data undergo both automated quality check and expert review for metadata consistency and data quality. All database and data handling tools are built in the open-access R computational language, and an official ISRaD code library is available through the R library repository, CRAN. All code and data are available in an open Git repository (Beem-Miller

et al., 2021). New functions and explanatory vignettes can be submitted by users for inclusion in the R package. The project website contains information, links, guides, and updates on the project (ISRaD, 2018-2021).

## A3 Coastal Carbon Research Coordination Network

The Coastal Carbon Research Coordination Network (CCRCN) was formed to accelerate the pace of discovery in coastal wetland carbon science by providing the community with access to data, analysis tools, and synthesis opportunities. Funded by a National Science Foundation Research Coordination Network, the project's primary staff includes a funded research scientist as well as several part-time data technicians. Besides organizing topical working groups and communities events, one of the primary engagements of the CCRCN is the development and maintenance of a database of carbon stock and sequestration in coastal marshes, mangroves, swamps, scrub/shrub, and seagrass.

Both the database and its software is hosted on GitHub (Holmquist et al., 2018 - 2021), and its structure and naming conventions (Holmquist, 2018) were based upon a series of iterative conversations with a range of experts. Integration of new datasets into the database is via scripted transcription, by which curation to CCRCN standards is performed in a unique "hook" script tailored for each dataset. A suite of helper tools aids unit conversion, quality control, and spatiotemporal processing specific to soil carbon data. Datasets are joined together to construct the multi-level database, partitioned primarily by scale of observation (depth series, core/plot, site, and methods levels). An automatically-generated bibliography tracks primary citations of data contributors alongside the secondary citation of the database itself. Post synthesis QA/QC script identifies possible duplicate plot-level entries between datasets. Internally-facing visuals and reports are generated via Markdown implementation to track database growth as well as geographic/biophysical gaps in the database. Finally, the online version of the database feeds into the backend of its primary public interface, the Coastal Carbon Atlas (Holmquist, 2021). This R Shiny App allows anyone to explore global representation, as well as query desired according to a variety of environmental/methodological parameters, then download the data and (importantly) the corresponding citations.

Parallel to synthesizing the database has been a concerted effort to generate data releases, each with its own DOI, as a service to data submitters as well as the coastal carbon community. This has included dedicated staff time towards outreach, formatting of datasets, generating metadata (based upon the Ecological Metadata Language standards) and assigning DOIs to datasets, which has so far resulted in the public release of 22 datasets the Smithsonian Figshare repository.

## A4 SoDaH

The Soils Data Harmonization (SoDaH) and Synthesis project features a tool suite for harmonizing soil organic matter data from disparate sources into a common data model, and a database of harmonized soil organic matter data and related variables that, as of this writing, includes data from over seventy unique studies (Wieder et al., 2020, 2021b). The product of a Long-Term Ecological Research (LTER) synthesis working group (LTER Soil Orgnaic matter Working Group, a), the project brought together soil scientists with diverse backgrounds and affiliations with scientific research networks to refine and evaluate theories of soil organic matter dynamics, and to produce a soil organic matter dataset that spans a wide range of environmental and experimental conditions.

SoDaH employs a keyed-translation approach using metadata about the data combined with conversion scripts to translate contributed data tables into the common data model. Metadata are organized at multiple levels, including around the study (e.g., study location, data provider) and data variables, which are subdivided into profile, layer, and fraction categories and based off of templates designed in the International Soil Carbon Network (Nave et al., 2016). Additional metadata fields facilitate the identification of experiment details and study design, allowing users to, for example, query data associated only with

specific manipulations or control conditions. The harmonization script (LTER Soil Orgnaic matter Working Group, b) maps user-provided metadata and data resulting in new flat file(s) in which the variable names and units, if relevant, are standardized in the output along with appropriate quality control. All output conforms to the specifications of the SoDaH data model thereby enabling the aggregation of output data from disparate studies into a single data file.

## A5 Data collection list

This is appendix is meant to be representative, not exhaustive.

1. SOils DAta Harmonization (SoDaH) & Synthesis (Wieder et al., 2020). Data available at https://lter.github.io/som-website/database.html (Wieder et al., 2021a). See Section A4

   – **Personnel**: 5 people over 4 yrs

   – **Description**: Layer-level soil data includes time series and experimental data.

– **Pipeline**: Aggregation of data from broad scale research networks. After level-0 data was contributed (both raw data +metadata template) SoDah supplies a script. First harmonization of raw data into a common format takes place and then it is aggregated into an a flat csv file.

   – **Size**: 303.6 MB

2. Soil warming meta-anlaysis (Crowther et al., 2016). Data available on request. See Section A1

– **Personnel**: 2 data aggregators with multitude of contributors and coauthors over 4 yrs

   – **Description**: Soil C in ambient and warmed soils. Field soil warming experiments

   – **Pipeline**: Used a template approach. The data template only asked for a brief description of the site location (latitude, longitude, etc.), the measurement they are looking for in their project, and how the author collected that measurement. This very restricted data model is then much easier (and more satisfying) to collaborate with data

products to fill out.

   – **Size**: 114 KB

3. vanGestel - field treatments van Gestel et al. (2018). Data available in supplementary information and raw data available on request

   – **Personnel**: 2 over 3 years

– **Description**: Soil C in ambient and warmed soils. Field treatments and soil warming

     – **Pipeline**: Excel was used as the repository for raw data (no unit conversion, no gap-filling etc.), with different sheets comprised of different aspects of the C cycle (soil C, plant C etc), and 1 sheet including meta-data. All computations/harmonization/QA/QC etc. done in R.

     – **Size**: 1 MB

4. World Soil Information Service (WoSIS) (Batjes et al., 2020). Data available at https://www.isric.org/explore/wosis (Batjes and Calisto).

     – **Personnel**: Institutional, 20 years and ongoing

     – **Description**: Focuses on bulk soil characteristics. To serve the user with a selection of standardised/ harmonised soil profile data. These quality-assessed data may be used to underpin digital soil mapping and a range of global
assessments.

     – **Pipeline**: Submitted data is first preserved in the ISRIC WDC-Soils Data Repository. It is then quality-assessed, standardised, and imported into the WoSIS data model itself. It is harmonized where possible using consistent procedures.

     – **Size**: Varies based on portion of database

5. SoilHealthDB (Jian et al., 2020). Data available at https://github.com/jinshijian/SoilHealthDB (Jian et al.).

     – **Personnel**: 3 people, initial publication 2020 with ongoing data contributions

     – **Description**: 42 soil health indicators and 45 background indicators. The primary goal is to enable the research community to perform comprehensive analyses of soil health changes related to cropland conservation management.

– **Pipeline**: Data extracted to Excel template; basic scripted QA/QC scripts; no unit conversion/harmonization (leaves this to future users); database and code available on github

     – **Size**: 64MB including metadata and documentation

6. International Soil Carbon Network 3 (ISCN3). Data available at https://iscn.fluxdata.org/data/ (ISC).

     – **Personnel**: 5 leads over 4 years to first publication and ongoing

– **Description**: Soil C stocks and associated site and bulk soil characteristics. To provide a science-based network that facilitates data sharing, assembles databases, identifies gaps in data coverage, and enables spatially explicit assessments of soil carbon in context of landscape, climate, land use, and biotic variables

     – **Pipeline**:SQL-based ingest of Excel templates or Access tables, hybrid scripted-manual QA/QC, rules-based computation of soil C stocks

– **Size**: 300 Mb

7. Soil Incubation Database (SIDb) (Schädel et al., 2020). Data available at https://soilbgc-datashare.github.io/sidb/ (Sierra).

      – **Personnel**: 8 people over 4 years to first publication and ongoing

      – **Description**: Soil incubated $CO_2$ and $CO_2$ isotope data along with ancillary variables. Time series data of soil incubations and soil respiration.

– **Pipeline**: Templates for data in csv format and metadata in ymal format. The data is processed by an R package that checks for consistency between data and metadata. The R package also provides functions to access and manipulate data.

      – **Size**: 1 MB

8. Coastal Carbon Research Coordination Network (CC-RCN). Data available at https://serc.si.edu/coastalcarbon/about
505 (Holmquist, 2018). See Section A3.

      – **Personnel**: 3 PIs, 5 steering group members, 2 staff over 5 yrs

      – **Description**: Tidal wetland soil C stocks and ancillary information. Provide access to data, analysis tools and synthesis products to accelerate the pace of discovery in coastal wetland C science.

      – **Pipeline**: Once the data is attained, the technician codes out the methods metadata, then does an initial data inspec-
510 tion by going through one or two drafts of a data release, and finally uploads it to Figshare. Custom R-functions and hook scripts are used for preparing the actual data releases. They do things like take quick looks at the range of the datasets to make sure that everything is how it should be. When preparing data releases, they prepare 'EML' style metadata.

      – **Size**: 14.3 MB

9. International Soil Radiocarbon Database (ISRaD) (Lawrence et al., 2020). Data available at https://soilradiocarbon.org/ (Beem-Miller et al., 2021). See Section A2.

      – **Personnel**: 30+ people, 6 yrs to first publication and ongoing

      – **Description**: Soil physicochemical data, isotopic data, fractionation data, incubation data. Aggregation and harmonization of radiocarbon and fraction data from previously published work.

– **Pipeline**: Standardized Excel template; scripted QA/QC; github repository

      – **Size**: 31 MB

10. SoilTemp (Lembrechts et al., 2020). Data available at https://soiltemp.weebly.com/ (Soi)

      – **Personnel**: Core team of 5-10 developers, 100s of data contributors. 5 yrs and ongoing

- **Description**: In-situ soil temperature and other microclimate time series, soil surface vegetation characteristics. Building a global database of soil and near-surface temperatures, and associated species occurrence data for use in ecological modelling.
- **Pipeline**: Provides a CSV-format template to data users, and tools in development to facilitate submission via an R package. Some curation and harmonization is performed via a scripted approach, although the workflow is in active development.
- **Size**: 30 GB

## Appendix B: Future Dream

It is often difficult to envision how inter-connected soil data could impact science. Here we have taken the liberty of a bit of creative writing to envision what the future of soil science could be in a data connected world.

### B1  A new data-savvy world

In the late 21st century, soil data are effortlessly collected and collated. There are still divisions between groups on what exactly the correct way to fill in certain missing data is, but in general most instruments come with their own connection interfaces that ensure interoperability between sensor data. It took the field scientists a little longer to get behind standardized digital records but most researchers now collaborate with their data libraries and archivists to adhere to established data standards.

Occasionally a researcher will design a completely novel method or experimental treatment that will require additions or modifications to soil semantic tools. These cases are highly sought after by informaticians and extending an existing international standard has been known to launch the careers of young researchers. More often, researchers will review new data archives to vet their annotations and report this as they do in manuscript reviews. Researchers of course complain about this additional workload, but recognize that if they are going to generate data, then there is an obligation to review their colleagues' data. Once or twice in their career a researcher will serve as a domain expert on a relevant ontology board, providing updates and revisions to these key international resources.

Digital data archives are now entirely annotated and the idea of putting data online without reference to an international ontology is considered irresponsible. Data are annotated with one or more ontologies from different sub-domains and there are a range of AI/ML tools that can leverage these annotations to create an integrated database. Model development and meta-analysis studies now spend the bulk of their time honing hypotheses instead of cleaning data.

The great data rescue projects of decades past are sadly, mostly done now. An entire generation of researchers cut their teeth combing through old paper archives and fighting with optical recognition characters. New researchers lament that this highly fruitful line of 'new' research data is now mostly spent. Designing a new sensor processing pipeline just isn't as romantic as speculating on the nature of that old coffee stained field journal.

Contrary to popular belief, the 'traditional' skills of soil observation (hand texturing soils, matching horizon colors) are more in demand than ever. The ongoing climate crisis has now defined several generations of researchers and reignited interest

in soils beyond an agricultural context. Knowing your soil and how you impact that soil is as important as water quality, soil science is a fundamental curriculum element for not only foresters and conservation majors but also urban planners and backyard gardeners. New passionate generations of young students have grown up on soil judging competitions and soil reports are common for any land or home purchase.

## B2   The era of big data rescue

In the middle of the 21st century, we are in the heyday of the Big Data Rescue of the 21st century. Driven by the need to understand the impact of legacy management and new data scrubbing technologies, researchers have dived deep into the filing cabinets and paper archives of the past century. In addition to the traditional literature review, new graduate students now do targeted data rescue chapters as part of their dissertation. These data rescue projects have also drawn new researchers from the library, data, and other sciences into soils and is also used as a common undergraduate research project.

Corporations and governments increasingly recognize the value of legacy datasets and actively participate in disseminating data to the public, though redaction of sensitive information is necessary. While the impact of 'good/bad' data metrics on land valuation are still a concern, double-blind methods are being developed to increase the security of data sharing and automate the obfuscation of sensitive data.

Ontologies and other semantic resources are increasingly being adapted and extended by domain scientists. Unfortunately, there are several competing standards reflecting national, domain, and general political divisions in the research community. However there are several clearly identified mature semantic resources that most disciplines agree are pretty good. Data management plans from funders now require identifying semantic resources in addition to the final data archive.

## B3   A post-pandemic, better-connected world

Over the next few years, the soils community has fully recognized that we have a data problem, that is actually a community problem. The collecting and publishing science in isolated labs has become increasingly frustrating to new researchers used to instantaneous web results. The COVID-19 Pandemic forced a rapid shift in how science is done, moving what might have been a few day workshop into a longer slow-burn virtual collaboration over months. This led to a new kind of decentralized project management where most projects are now interconnected to similar researchers in regular virtual seminars and working groups.

This increase in researcher interactions has led to an increase in data interactions. As research interact more online, there was a correlating increase in comparing data from their study with their colleagues results. This led to an informal common vocabulary and data methodologies that increasingly showed up on newly archived data. Some graduate students are starting to dive into data rescue operations and further expanding these vocabularies to include older methodologies. Combining automated optical character recognition of scanned documents with manual corrections, these older data are providing valuable insights into climate change.

*Author contributions.* All authors contributed to the writing on of this manuscript. Facilitator and organizational lead was K Todd-Brown.

*Competing interests.* The authors declare no competing interests.

*Acknowledgements.* We would like to thank several anonymous survey respondents for their contributions.

This work is based on materials provided by the ESIP Lab with support from the National Aeronautics and Space Administration (NASA), National Oceanic and Atmospheric Administration (NOAA) and the United States Geologic Survey (USGS).

James Holmquist was funded by the Coastal Carbon Research Coordination Network (DEB-1655622). Natasja van Gestel was funded by the Department of Energy's Biological and Environmental Research (DOE-BER) program (DE-SC0010632). Rose Abramoff was funded by the European Union's Horizon 2020 research and innovation programme under the Marie Sklodowska-Curie grant agreement No 834169 and the United States Department of Energy, Office of Science, Office of Biological and Environmental Research. Oak Ridge National Laboratory is managed by UT-Battelle, LLC, for the United States Department of Energy under contract DE-AC05-00OR22725.

We would like to thank Christina Schädel (Center for Ecosystem Science and Society, Northern Arizona University), Tom Crowther (ETH Zurich), Adrien C. Finzi (Boston University), Karis McFarlane (Lawrence Livermore National Laboratory), Corey R. Lawrence (USGS), Chelsea Carey (Point Blue Conservation Science), Gabriel Reuben Smith (Stanford University; ETH Zurich) for their contribution to conversations on this topic.

The findings and conclusions in this publication are those of the authors and should not be construed to represent any official USDA or U.S. Government determination or policy.

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
