# Peer review of "Notice: This manuscript has been authored by UT-Battelle, LLC, under contract DE-AC05-00OR22725 with the US Department of Energy (DOE). The US government retains and the publisher, by accepting the article for publication, acknowledges that the US government retains a nonexclusive, paid-up, irrevoca"

_Biogeosciences, 2021_

## Author Response (AR1)

**Round 1 reply to reviews**

Link to discussion: https://bg.copernicus.org/preprints/bg-2021-323/#discussion

**Instructions**: Please make new edits in suggestion mode. If you see a suggested edit that you like, please accept the edit. Please refer to potential changes in the manuscript by line number. Internal snarky comments or points for discussion are encouraged in the comments.

**DOE review:**

Please add the following notice to the submission to the journal so that intent to archive is communicated.

" Notice: This manuscript has been authored by UT-Battelle, LLC, under contract DE-AC05-00OR22725 with the US Department of Energy (DOE). The US government retains and the publisher, by accepting the article for publication, acknowledges that the US government retains a nonexclusive, paid-up, irrevocable, worldwide license to publish or reproduce the published form of this manuscript, or allow others to do so, for US government purposes. DOE will provide public access to these results of federally sponsored research in accordance with the DOE Public Access Plan (http://energy.gov/downloads/doe-public-access-plan). "

**Forest Service review:**

*The Forest service requested the following passage be revised due to concerns regarding portrayal of government entities as ethically questionable.*
    *"Big corporations and governments have taken on the task of parcelling out data to the general public, stripping out sensitive information reduces the utility of the data but is seen as a necessary evil. Ethicists are still debating whether this gives the corporations/governments too much knowledge, and double-blind methods are being developed to obfuscate sensitive data even from the data holders."*

The following passage has been proposed as a replacement:
    "Corporations and governments increasingly recognize the value of legacy datasets and actively participate in disseminating data to the public, though redaction of sensitive information is necessary. Double-blind methods are being developed to increase the security of data sharing and automate the obfuscation of sensitive data."

**Reviewer comment 1**

We would like to thank the reviewer for their comments and note that the original review is in *italic* with our reply following.

*General comments*

*This narrative/opinion manuscript describes the promise and challenges of soil databases, data discovery and harmonization, and related efforts. This is a topical and timely topic, given the many top-down and bottom-up efforts that have sprung up in this area over the last 10-15 years. The manuscript is reasonably well written and has interesting points, but I think there are some significant weaknesses here as well.*

*First, the authors seem to repeatedly conflate and/or mix up "big" and "open" data—starting with the title, see #1 below. They're obviously not the same thing, and most of the ms seems to actually focus on \*open\* data. It would be good to clearly define these terms, their distinctions, and use the terminology consistently and correctly throughout.*

The authors respect that there could be some confusion for readers when it comes to how we are referring to "Big Data" within the manuscript. We believe that 'big' extends beyond the absolute size of the data files and includes 'diverse' data as well; we now pair these two words consistently throughout the paper, including in the title. We also disagree that our manuscript focuses on "open" data, although it is understandable that the reviewer came away with this impression. All of the strategies discussed in this paper could be applied to 'closed' or proprietary data, although we acknowledge additional difficulties with proprietary data in section 3.1. We suggest the following changes that will hopefully help clarify this point.

- A new title: **"The promise of big diverse soil data…"**
- Abstract ln 2: **"In the age of big data, soil data are more available and richer than ever…"**
- Ln 50: **It's important to note here that FAIR does not always mean open freely reusable data. Indeed the FAIR Data motto makes this difference quite clear: "as open as possible, as closed as necessary", and this becomes particularly important for data that has possible economic impact (Luque 2019).**
  - (Luque, C., 2019. Open data and FAIR data: Differences and similarities. Plataforma OGoov [en ligne], 23.; "Open Data". (2019), *OGoov Open Government Platform*, 6 May, available at: https://www.ogoov.com/en/glosario/open-data/ (accessed 6 March 2022).; and "la Investigación", B.U.A. al A. y. (2017), "Biblioguías: Datos de investigación: gestión, datos abiertos (Open Data): INTRODUCCION", available at: https://biblioguias.unex.es/datos_de_investigacion (accessed 6 March 2022).)

- Ln 183: "**To reach this hope it's important not just to work with large volumes of data but also diverse observation and measurements, and do so in a way that is trusted.**"

Thank you for pointing this out to us and giving us an opportunity to improve the paper.

*Second, what about this is unique to soils? I struggled to find anything in the ms that wouldn't apply to environmental data more generally, and thus what points are made here that haven't already been made by authors like Wolkovich 2012 (http://dx.doi.org/10.1111/j.1365-2486.2012.02693.x) or Rüegg et al. 2014 (http://dx.doi.org/10.1890/120375). What exactly is the value added here, in other words? That's not as clear as it needs to be.*

Both reviewers bring up this point that soils data needs are not unique. And we agree that the strategies for creating and maintaining soil databases are not unique. We have modified the introduction and conclusion to highlight the relevance of these findings to the broader environmental community. Soils are unique in their importance and societal relevance but not in these identified data challenges. This paper strives to lay out how soil scientists currently aggregate data and point out various strengths and weaknesses of this approach. It is, by design, experiential and written by soil scientists not informaticians (see lines 59-61). **Specifically we proposed adding on ln 60: "The approach and issues outlined in this paper are undoubtedly not unique to soils and are relevant to a wide range of scientific data, particularly environmental data. However we present this as a case study of soil specific database construction."**

Wolkovich etal 2012 contends that the problem is motivation and knowledge on the part of the data collectors/providers; which we disagree with, data providers are often highly motivated to see their data have as broad and large an impact as possible. **We will add the following statement in the introduction on line 50. "Indeed previous research has identified challenges with educating and motivating data providers to publish their data sets (Wolkovich et al 2012)."**

Ruegg etal 2014 does come to similar conclusions that a common descriptive framework would benefit the field more broadly. However we show how researchers are currently conducting data harmonization without such resources in this paper and that folks will generally find this template approach both too large and too small. In addition, we would contend that their suggestion of informatics being an early part of the project design is not tractable for most small, single PI data and demonstrate with this manuscript that

researchers are moving the field forward without this element. **We will add Ruegg etal 2014 as an example of a suggested standard approach on line 240.**

*Third, there are many curious omissions from the references, I thought. For example, Crystal-Ornelas et al. 2021 ("A guide to using GitHub for developing and versioning data standards and reporting formats", Earth Space Sci., https://doi.org/10.1029/2021ea001797) is relevant in many places.*

Thank you for bringing Crystal-Ornelas 2021 to our attention, we've added this to the recommended best practices for community semantic tool development. **Specifically we propose adding to ln 245: "...practices from the open source community such as version control also are a critical tool (Crystal-Ornelas et al 2021)"**

*Re reporting formats, Bond-Lamberty et al. 2021 ("A reporting format for field measurements of soil respiration", Ecol. Inform., 62, 101280), which was part of a special issue on integrating long-tail data: https://www.sciencedirect.com/journal/ecological-informatics/special-issue/101T38RSLSF.*

Again, thank you for bringing this 2021 paper to our attention and **we have added it as a contrasting example of data standards on line 239: "...(for example, the format suggested by Bond-Lamberty et al 2021")**. However we will point out that one of our main findings was that data standards as described here are insufficient due to the diversity of measurements and study design.

*In a related vein, the SRDB (https://github.com/bpbond/srdb) is a decade older than most of the efforts discussed here and widely used and cited, so might be worth a mention as well, unless you're particularly focusing on stocks but not fluxes.*

We agree that SRDB is an excellent example of soil data harmonization, and further add that the Worldwide soil carbon and nitrogen data  Zinke et al 1986 is an even older example of soil data harmonization. We've added this to our introduction and included a table of active/recent soil database projects. **We are removing the ILAMB sections (ln 41-45) and replacing this with the following: "A number of databases have been compiled in soils data around specific themes or measurement types including: soil carbon and nitrogen (Worldwide soil carbon and nitrogen data  Zinke et al 1986; International Soil Carbon Network database ISCN Nave 2015), field based soil respiration (Soil Respiration Data base; Bond-Lamberty and Thomson 2010, Jian et al  2021), lab-based heterotrophic respiration (Soil Incubation Database), soil radiocarbon (International Soil Radiocarbon Database), and coastal soils (Coastal Carbon Research Coordination Network Database) (See Table XX for a complete list with database properties)."**

*Unlike most of the other efforts discussed, SoilGrids (Hengl et al.) really is big data (pretty big anyway) and that should be noted.*

SoilGrids is an excellent collection of data products that highlights how we differentiate between databases and data products in this paper. We address this, and related comments from R2, beginning on line 45.

Suggested text:

**Soil resources curated by ISRIC (https://www.isric.org/) provide another example of how soil data feed into larger products. After archival on ISRIC servers, datasets from individual providers are incorporated into the World Soil Information Service workflow (WoSIS; https://www.isric.org/explore/wosis). The WoSIS workflow includes mapping diverse data contributions to a standard data model, harmonization, and distribution. Distribution includes a database, as defined in this paper (the WoSIS Soil Profile Database; https://www.isric.org/explore/wosis/faq-wosis#How_should_the_WoSIS_datasets_be _cited?), as well as derived data products, such as SoilGrids (Hengl T, de Jesus JM, MacMillan RA, Batjes NH, Heuvelink GBM, et al. (2014) SoilGrids1km — Global Soil Information Based on Automated Mapping. PLoS ONE 9(8): e105992. doi:10.1371/journal.pone.0105992)**

*In summary, there are many points of interest here, and I applaud this effort by the authors. The current ms has some significant issues, and would benefit from tighter language—it's pretty long—and clearer novelty.*

We highlighted the generality and novelty of the soil centered nature of this paper in the above change to the introduction. We tightened the language, for example, by removing the ILAMB reference in the introduction and replacing it with a review of a selection of current soils databases. We hope that this addressed your concerns.

*Specific comments*

Most of these were specifically addressed above.

1. *Title: a bit odd (most of this manuscript is about \*open\* data, not \*big\* data), and it's a run-on sentence; consider rewording*

We hope that our clarification in the introduction addressed this wording and suggest revising the title to include "big diverse soil data".

2. *Line 182: do you mean "open" data here? That's not what big data is*

See above.

3. *193: …just like any other environmental data*

See above.

4. *212: wow, that (60%) is shocking*

We know right?!?!

5. *215-: do you mean "time" of collection, i.e. 1400 hours? Or "date"?*

Good point. **We suggest adding the following to ln 217: "While the exact resolution will vary depending on the measurement (for example, parent material may just need the decade of collection while soil respiration may need a minute resolution), recording [...]"**

6. *239: see recent ESS-DIVE -funded papers on data standards/reporting formats in Ecological Informatics*

See above.

7. *290: a better analogy might be the \*software\* review process? See Crystal-Ornelas paper*

See above.

8. *296-312: this is all restating material above, should be removed*

We respectfully feel that this ties the ending of the paper back to the introduction and suggest keeping this section.

9. *403: haha, data, singular or plural? Both!*

Clearly! Thank you for the catch. We've replaced this with "**data are**"

10. *433: what is this referencing? Confusing*

We've added the following leading sentence for this paragraph on ln 433: **Data privacy concerns and the impact of 'good/bad' data metrics on land valuation are still an issue but "trusted' data holders are attempting to address this.**

**Reviewer comment 2**

*This is solid, carefully written, succinct perspective on the landscape of soil data and harmonization efforts. I commend the authors for putting together clear, thoughtful state of the science.*

 Thank you!

*This said, I would also encourage the authors to think a bit more broadly about the context in which they are laying out these suggestions. Specifically, with the rapidly evolving soil C sequestration landscape and an infusion of private interests into soil C world (e.g. https://seqana.com/, IndigoAg, etc), how should academics, industry, ngo's and government agencies maintain data access and communication in what's potentially a more crowded, active (and presumably better funded) field? I appreciate this touches on ideas that are broader and somewhat more existential than the soils data challenge the paper more narrowly addresses- but it*

*seems relevant to contextualize the broader landscape of who and why harmonize soils data-
beyond the how it can be done better.*

Excellent point here. We will add additional comments on recent interests on soil carbon
sequestration to the introduction, but would like to point out that this is an entire topic of study
itself. Specifically, we will remove lns 43-45 and add a brief discussion on carbon sequestration
after line 237 in the discussion.

**The uses of soil databases for research context are varied (for example Earth system
model benchmarking Collier et al 2018) but there are other private economic impacts of
having soil data available. Soil health metrics in public databases could impact land
evaluation and there is increasing interest in soil carbon data from carbon markets for
offsetting CO2 emissions. As mentioned in the geolocation section, specific information on
the nutrient and water retention of a soil can make it more or less valuable, making
landowners reluctant to release data. More recently, an increasing interest in generating
carbon offsets by increasing soil sequestration has led to a proliferation of new venture
corporations that either generate new or use available soil data in order to define land
management practices to increase soil C stocks (e.g. IndigoAg, CIBO Technologies,
Seqana, Regrow, Nori, LoamBio). Industry companies generally treat data that they collect
or process as part of their intellectual property, which is kept private. While there is clearly
scientific value in these data, it's unclear how researchers, landholders, and private
companies will negotiate the use and integration of these data into research outputs.
Nonetheless, privately held data would also benefit from connecting with community
developed standards.**

*My remaining comments are relatively minor, and largely intended to clarify aspects of the text.*

*I'm not sure I agree with the statement in Line 43-45. Modeled soil properties (here I'm thinking of
hydraulic and thermal properties) rely on pedotransfer functions that use input data of soil physical
characteristics (texture and organic matter content). None of these 'soil properties' are used for
benchmarking or evaluation, making me wonder what the growing need for more data are really
needed for- especially if ILAMB already uses information on soil C stocks and inferred turnover
times?*

We can see how this was confusing, this was meant to refer to carbon and nutrient stocks but on
review this section is unclear. **We are removing the ILAMB sections (ln 41-45) and
replacing this with the following: "A number of databases have been compiled in soils
data around specific themes or measurement types including: soil carbon and nitrogen
(Worldwide soil carbon and nitrogen data  Zinke et al 1986; International Soil Carbon
Network database ISCN Nave 2015), field based soil respiration (Soil Respiration Data
base; Bond-Lamberty and Thomson 2010, Jian et al  2021), lab-base heterotrophic
respiration (Soil Incubation Database), soil radiocarbon (International Soil Radiocarbon**

**Database), and coastal soils (Coastal Carbon Research Coordination Network Database) (See Table XX for a complete list with database creation strategies)."**

*Moreover, data products like SoilGrids already exist, which seems to have a wealth of data that can be used as inputs for or evaluation of Earth system models. Are you suggesting new efforts should go into recreating or augmenting the data processing wheel that informs ISRIC data products (SoilGrids and the Harmonized World Soils Database)? I don't get the sense this is what the authors are envisioning? I also appreciate that "This is just one of many potential uses for harmonized soil data", but I do worry that as written the authors are implying that the harmonized datasets we do have somehow do not reflect FAIR principles.*

We contend that soil data products (like SoilGrids and HWSD) are not the same as an aggregated soil database and that a soil database is necessary to generate these data products but has other use cases as well. We address this, and related comments from R1, beginning on line 45.

Suggested text:

**Soil resources curated by ISRIC (https://www.isric.org/) provide another example of how soil data feed into larger products. After archival on ISRIC servers, datasets from individual providers are incorporated into the World Soil Information Service workflow (WoSIS; https://www.isric.org/explore/wosis). The WoSIS workflow includes mapping diverse data contributions to a standard data model, harmonization, and distribution. Distribution includes a database, as defined in this paper (the WoSIS Soil Profile Database; https://www.isric.org/explore/wosis/faq-wosis#How_should_the_WoSIS_datasets_be _cited?), as well as derived data products, such as SoilGrids (Hengl T, de Jesus JM, MacMillan RA, Batjes NH, Heuvelink GBM, et al. (2014) SoilGrids1km — Global Soil Information Based on Automated Mapping. PLoS ONE 9(8): e105992. doi:10.1371/journal.pone.0105992)**

*I really like the tone of the last paragraph of the introduction, which seems constructive and positive.*

Thank you! We are excited about the direction of the field!

*I also like the preview for what's ahead in in section 2 (lines 73-74) and wonder if the subheadings for section two and headings in Fig 1 should use identical language (acquisition, harmonization, curation, and publication.*

Agreed. **We will integrate this language on ln 73-74 into the figure and headings.**

*Line 79-84, I appreciate the challenge you're trying to articulate- but it kinds of seems like you're suggesting reviewers or journals need better evaluation of data publishing standards. I wonder I*

*this is really where the responsibility should lie, specifically because I don't think as a community, we're well trained in best management of data practices.*

Good point. We did not intend for the responsibility to lie with peer-reviewed journal, rather we diverted the focus to one that highlights that challenge as to who would be responsible, so it is more of an open question. **We will add the following to ln 81 "... "high standard" are and whom is responsible for ensuring these standards are met. To complicate matters, key…"**

*I think given better information, data providers would happily provide more useful datasets to repositories, but don't know how. Maybe this is what's implied in line 83 with data providers who 'become frustrated'? I realize you're trying to be brief here- and maybe a solution is articulated in Section 3- but I do worry that the takeaway message from this paragraph is 'currently archived data are incomplete and therefore useless, and we're not really going to tell you how to make them better'.*

Good point. **We propose extending this paragraph and adding to ln 84 "This is not to say that archiving data for the purpose of meeting funder requirements or reproducing the associated analysis can not be useful in and of itself. However this does not automatically lend the data to integration in a database."**

*Line 87, what's a harmonized template?*

We agree this is unclear - we will reframe as **'aggregator provided template'**

*Line 99  What are TRUST and CARE?  If an aim of this manuscript is to broadly educate soil-minded scientists on best data practices describing features of these practices should be briefly articulated (not just referenced).*

We propose adding to line 97: **In general however, we feel that direct collaboration between data providers and data aggregators is a critical relationship to nurture. Other critical relationships for good data governance have been articulated by recent extensions of the FAIR Principles, including TRUST and CARE.  The TRUST Principles (Transparency; Responsibility; User focus; Sustainability; Technology) articulate key features for trustworthy digital repositories, which are essential for preserving data access and reuse over time (Lin et al., 2020).  The CARE Principles for Indigenous Data Governance (Collective benefit, Authority to control; Responsibility; Ethics) position decisions related to data management and reuse in the context of Indigenous cultures and knowledge systems, highlighting actions that ultimately support Indigenous data sovereignty (Carroll et al., 2020). As the community continues to converge on shared tenets of good data governance, it is becoming increasingly clear that "just put it in a repository" is only the beginning.**

*Line 105.  These different transcription / translation methods are nicely described in the text, with examples in Appendix A.  Would a table help emphasize similarities and differences of databases*

*listed in Appendix A? Building on this table idea, I don't think Avni's 2019 paper really provides much depth on these features of the databases. It seems like some A2-A4 all have some high-level similarities- e.g. R code provided on github. A3 and A4 both have a Shiny App (although not listed for A4). Does ISRAD have a shiny app too, or just it's own R library? For what it's worth I feel like some of these back end usability features are helpful if we want people to engage with the harmonized datasets.*

Capturing these differences in a table form was challenging which is why we went with the narrative structure however **we will create a table capturing some of these database strategies and add it as a new table.**

*Finally, both ISRAD and SODAH were organized with the nested hierarchy established with ISCN. Should this be mentioned? Should ISCN be highlighted in the text (a number of co-authors have contributed to this effort)? This hierarchical organization of the data is implied, but maybe not explicitly established in the metadata and data models we are or should be using.*

Good point, **We will add the ISCN connection to these two project descriptions.**

*Section 2.2, It seems like scripted transcription requires clear dictionaries, vocabulary and metadata to be successful, but based on text in 2.1 this is not common, OR is this just happening in keyed translation?*

Both manual transcription and scripted methods require clear metadata descriptions that are formalized in different ways. We'll add this point here on lins 129: **While this approach has the most explicit need for clear semantic resources, these are also essential for creating effective manual transcription templates and protocols.**

*Section 2.3 is pretty brief Would additional examples be helpful here to illustrate how different efforts have gapfilled or pruned their data? How do these databases expand- which seems important aspect of curation (although discussed in 2.4 for COSORE).*

With respect, these strategies are extremely diverse and beyond the scope of this paper, see lines 148-149.

*Line 275, I may add something aboveground to this list (as vegetation, land use, productivity and climate are also important for belowground measurements, but rarely co-located with belowground measurements being collected).*

We will extend section 2.3 to talk about annotation of soil observations with aboveground data (ie ISRaD annotating mean annual temperature and precipitation) Specifically ln 147 **These activities include expanding the environmental context for a particular soil; for example, extracting net primary productivity and land use classification from satellite products.**

*Section 3.2 (or in the introduction). Are there successful examples we can learn from elsewhere (e.g. fluxnet, TRY, or FRED) how can these other database models be translated for soils? What unique challenges do the landscape of soils data provide?*

Soils are not unique and many of these are broad challenges in environmental data. **Specifically we proposed adding on ln 60: "The approach and issues outlined in this paper are undoubtedly not unique to soils and are relevant to a wide range of scientific data, particularly environmental data. However we present this as a case study of soil specific database construction."**

*I'm 100% behind the suggestions and vision the authors laid out, but I do wonder a bit about to what end? What are the pressing questions that a massive new soils database will let us address? Given the diversity of soil uses, measurements, and communities is a database of databases really what we need? OR, is the soil science community well enough served by individual collections of data that are more focused on more topical areas like radio carbon, respiration fluxes, spectral databases, or bulk C stocks? I realize this isn't you're grant proposal- but presumably it's heading that way.  The text clearly delineates data providers and data aggregators, but who are the data users that will ultimately do something with these datasets once they're wrangled into something more useful?*

You are correct of course that this paper focuses on data aggregators as a class rather than data users, we choose to do this because the user community is exceptionally diverse but data aggregation is a common activity across this group. Respectfully we choose to focus this paper on the how instead of the why.

*Finally, apologies on my delay in posting this review.*

*Citation: https://doi.org/10.5194/bg-2021-323-RC2*